# Past world economic production constrains current energy demands: Persistent scaling with implications for economic growth and climate change mitigation

Timothy J. Garrett[1]*, Matheus Grasselli[2], Stephen Keen[3]

**1** Department of Atmospheric Sciences, University of Utah, Salt Lake City, UT, United States of America, **2** Department of Mathematics, McMaster University, Hamilton, Ontario, Canada, **3** Institute for Strategy, Resilience and Security, University College London, London, United Kingdom

* tim.garrett@utah.edu

**Data Availability Statement:** All relevant data are within the manuscript and its Supporting Information files.

## Abstract

Climate change has become intertwined with the global economy. Here, we describe the contribution of inertia to future trends. Drawing from thermodynamic principles, and using 38 years of available statistics between 1980 to 2017, we find a constant scaling between current rates of world primary energy consumption $\mathcal{E}(t)$ and the historical time integral $W$ of past world inflation-adjusted economic production $Y$, or $W(t) = \int_0^t Y(t')dt'$. In each year, over a period during which both $\mathcal{E}$ and $W$ more than doubled, the ratio of the two remained nearly unchanged, that is $\lambda = \mathcal{E}(t)(t)/W(t) = 5.9 \pm 0.1$ Gigawatts per trillion 2010 US dollars. What this near constant implies is that current growth trends in energy consumption, population, and standard of living, perhaps counterintuitively, are determined by past innovations that have improved the economic production efficiency, or enabled use of less energy to transform raw materials into the makeup of civilization. Current observed growth rates agree well with predictions derived from available historical data. Future efforts to stabilize carbon dioxide emissions are likely also to be constrained by the contributions of past innovation to growth. Assuming no further efficiency gains, options look limited to rapid decarbonization of energy consumption through sustained implementation of at least one Gigawatt of renewable or nuclear power capacity per day. Alternatively, with continued reliance on fossil fuels, civilization could shift to a steady-state economy, one that devotes economic production exclusively to maintaining ongoing metabolic needs rather than to material expansion. Even if such actions could be achieved immediately, energy consumption would continue at its current level, and atmospheric carbon dioxide concentrations would only begin to balance natural sinks at concentrations exceeding 500 ppmv.

## 1 Thermodynamic overview of civilization growth

Like other biological systems [1, 2], the human economy interacts with its surroundings through flows of energy and matter [3, 4]. Collectively, we use primary energy resources to

**Funding:** This work was supported by the National Institute of Economic and Social Research to TJG, MG, and SK, whose views it does not represent.

**Competing interests:** The authors have declared that no competing interests exist.

power civilization and convert raw materials into the material make-up of civilization infra-structure or Earth's "technosphere", and in turn to further energy consumption growth [5, 6]. The circulations of our lives include the back-and-forth material exchange of people, goods, and information along transportation and communication networks, and our cardiovascular, pulmonary and nervous systems [7–10]. These require a power source, and any impulse of energy that passes through these networks is ultimately dissipated through frictional losses as waste heat. Concurrently, our infrastructure, bodies, and even our memories undergo decay. Without a continual drawdown of primary energy and raw material resources to continually rebuild, they inevitably falls apart.

So, while the entirety of humanity seems infinitely complex in its range of activities, what all social phenomena have in common is their need for energy. Yet, the sum total of their power consumption is constrained by the world total, or the rate of primary energy consumption $\mathcal{E}$. By consuming primary energy, for example through combustion, civilization is stretched to a potential $G$, higher than its equilibrium state with the environment characterized by $G = 0$ (Fig 1). This potential energy $G$ is converted into civilization circulations by doing reversible work $\mathcal{W}^{rev}$ to move civilization elements along pathways that align with the potential gradient $\nabla G$ [11] and, through frictional losses, work is converted to waste heat that is radiated to space. If the average velocity of the circulations is $\vec{v}$, and no primary energy is consumed, then the rate at which potential energy is dissipated is

$$\mathcal{W}^{rev} = -\frac{dG}{dt} = \vec{v} \cdot \nabla G = \frac{G}{\tau_d} \qquad (1)$$

where the dissipation time is $\tau_d = 1/(\vec{v} \cdot \nabla \ln G)$.

The reason civilization does not relax to an environmental equilibrium is that sustained primary energy consumption continually elevates $G$. For example, oil is extracted from a well for consumption, and there is back-pressure in the reserves to replenish supplies. Available statistics [12] suggest that civilization has been consuming energy nearly as fast as it has been produced. Since 1980, the mean ratio of global production to consumption has been 0.998 with an annual standard deviation smaller than 1%. So, while $G$ may fluctuate because consumption and dissipation are out of phase over rapid timescales shorter than $\tau_d$, averaged over timescales a bit longer, say one week, these variations are no longer apparent. Then, the quasi-equilibrium condition for civilization sustenance is:

$$\left(\frac{dG}{dt}\right)^{sust} = \mathcal{E}^{sust} - \frac{G}{\tau_d} \simeq 0 \qquad (2)$$

implying that $\mathcal{E}^{sust} = G/\tau_d$ is the energy consumption rate required to sustain civilization circulations, or in terms of the thermodynamic First Law, to maintain a short-term balance between external heating $\mathcal{E}^{sust}$ and doing reversible work $\mathcal{W}^{rev}$.

Civilization has a wide range of activities, each with their own energetic demands and time-scales that could be characterized by a power spectrum of consumption $\mathcal{E}_\tau^{sust}$ satisfying

$$\mathcal{E}^{sust} = \int_0^\infty \mathcal{E}_\tau^{sust} d\tau \qquad (3)$$

in which case the characteristic timescale for civilization is $\tau_d = \int_0^\infty \mathcal{E}_\tau^{sust} d\tau / \mathcal{E}^{sust}$. Given that approximately 40% of our time is spent alternating between our top two preferred locations, (e.g. home and work) [13], $\tau_d$ might be taken to be about 1 day. Currently, the world primary energy consumption rate is roughly 20 TW, so it follows from the quasi-equilibrium

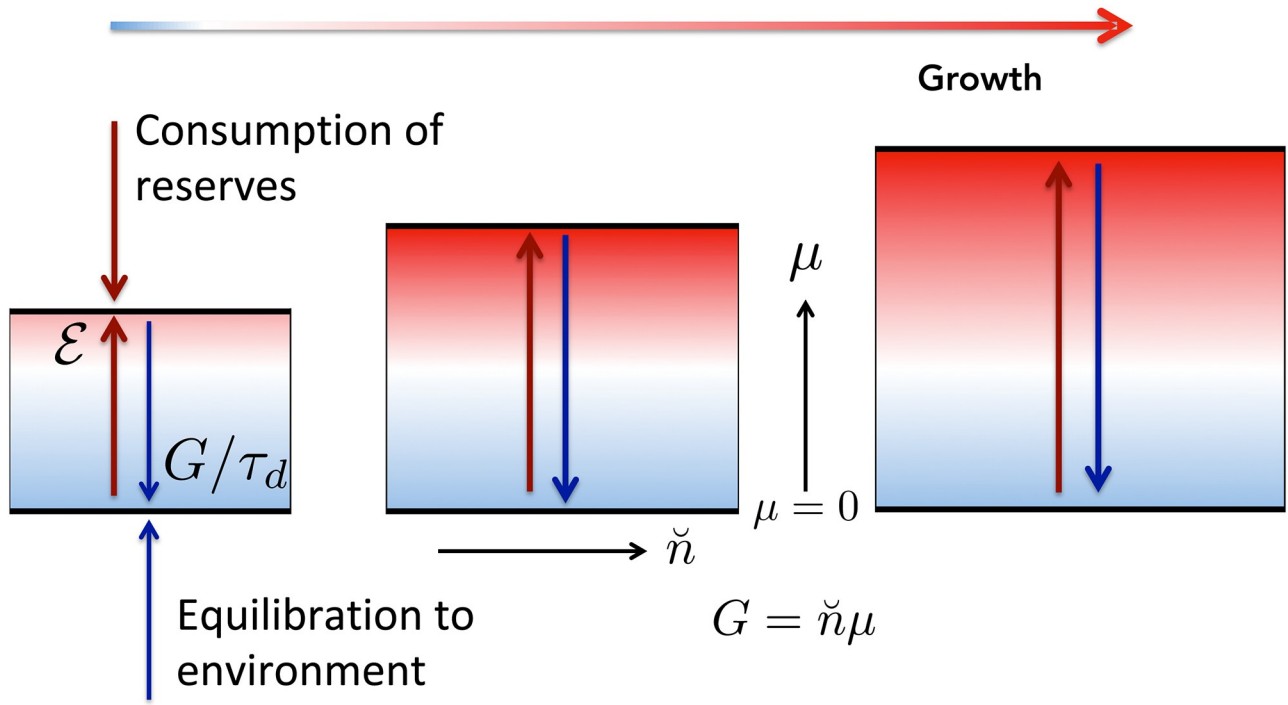

**Fig 1. Illustration of the short-term thermodynamic balance between primary energy consumption $\mathcal{E}$ and dissipation at rate $G/\tau_d$ (Eq 1) in a civilization experiencing long-term, proportionate material $\breve{n}$ and specific potential $\mu$ growth (Eq 14).**

relationship Eq (2) that

$$G = \mathcal{E}^{sust}\tau_d \simeq 20 \times 10^{12} \text{ J s}^{-1} \times 86400 \text{ s} = 1.7 \times 10^{18} \text{ J}, \tag{4}$$

equivalent to the energy contained in 270 million barrels of oil.

Obviously, such high potential is a far cry from where we started in the Stone Age. How did we become so strong? There has been no external hand to turn up the civilization flame. Purely mathematically, reaching our current state meant accumulating successive increments in potential $G$ over civilization's history:

$$G(t) = \int_0^t \frac{dG(t')}{dt'} dt' \tag{5}$$

Eq 2 implies that the primary energy supply $\mathcal{E}$ must have been greater than humanity's collective metabolic needs $\mathcal{E}^{sust}$ so that

$$\frac{dG}{dt} = \frac{G}{\tau_{long}} = \mathcal{E} - \mathcal{E}^{sust} > 0 \tag{6}$$

An important point here is that civilization growth timescales $\tau_{long}$ are decades to centuries, that is, much longer than $\tau_d$. For example, recent growth rates of global primary energy consumption are approximately 2% per year (or equivalent to $\tau_{long} \simeq 50$ years), so the implied difference between $\mathcal{E}$ and $\mathcal{E}^{sust}$ is only about 0.01%. Such growth might be imperceptible in our daily lives, but it does slowly accumulate. We can therefore write

$$\mathcal{E}^{sust} = (1 - \epsilon)\mathcal{E}, \tag{7}$$

where $\epsilon = \tau_d/(\tau_d + \tau_{long}) \ll 1$, so that the approximation $\mathcal{E} \simeq \mathcal{E}^{sust}$ can be made at any given time. For the remainder of $\mathcal{E}$, we have

$$\mathcal{W}^{irr} = \frac{dG}{dt} = \epsilon\mathcal{E}, \tag{8}$$

so that $\epsilon$ can be seen to be the efficiency of converting primary energy consumption $\mathcal{E}$ to the irreversible work $\mathcal{W}^{irr}$ that enables growth. Note how the sign on $\mathcal{W}^{irr}$ is opposite to that in Eq 1, where potential energy is dissipated to do the reversible work $\mathcal{W}^{rev}$ that maintains civilization circulations. Instead, irreversible work is done to grow the civilization potential $G$. Moreover, it follows that the growth rate of the potential is given by

$$\eta_G := \frac{d\ln G}{dt} = \frac{1}{G}\frac{dG}{dt} = \frac{1}{\tau_{long}} = \frac{\epsilon}{(1-\epsilon)\tau_d} \simeq \frac{\epsilon}{\tau_d} \tag{9}$$

Similarly, taking the derivative of

$$\mathcal{E} = \frac{\mathcal{E}^{sust}}{1-\epsilon} = \frac{G}{(1-\epsilon)\tau_d} \tag{10}$$

we find that the growth rate of primary energy consumption is

$$\eta_{\mathcal{E}} := \frac{d\ln\mathcal{E}}{dt} = \frac{1}{\mathcal{E}}\frac{d\mathcal{E}}{dt} = \frac{\epsilon}{(1-\epsilon)\tau_d}(1+\tau_d\eta_\epsilon) \simeq \frac{\epsilon}{\tau_d} \tag{11}$$

where

$$\eta_\epsilon := \frac{d\ln\epsilon}{dt} \tag{12}$$

is the growth rate of $\epsilon$ and $\tau_d \eta_\epsilon \ll 1$.

The mechanism by which an imbalance emerges between consumption and dissipation leading to $\mathcal{W}^{irr} > 0$ is not discussed in detail here. It can be argued to follow from civilization discovering accessible energy resources faster than they are consumed [14, 15]. The gradient $\nabla G$ becomes steeper nearer the energy source than to the dissipative sink, so that there is a net convergence of energy in civilization. Very generally, this process is analogous to the heat equation $(dG/dt)_{long} = \nabla \cdot (\mathcal{D}\nabla G) = \mathcal{D}\nabla^2 G$, where $\mathcal{D}$ is a constant diffusivity.

For more intuitive insight, consider a child as a more familiar complex system. From Eq 2, a child with potential $G$ has quasi-equilibrium metabolic needs of $\mathcal{E}^{sust} = G/\tau_d$—say about 50 Watts, or about 500 kJ per day per kg. Any food energy the child consumes is used to do reversible work $\mathcal{W}^{rev}$ to maintain rapid internal neurological, respiratory, cardiovascular, and nervous circulations and to reconstitute food nutrients into its material makeup. If the energy in the proteins, carbohydrates and fats of accessible food is only just sufficient to offset decay and keep the child alive, then the child is at steady-state and $G$ and $\mathcal{E}^{sust}$ do not change. A healthy child, however, does net irreversible work at rate $\mathcal{W}^{irr} = \epsilon\mathcal{E}$ to convert some small portion of the energy in food into accumulation of body matter through a conversion factor of about 30 MJ kg$^{-1}$. The child eventually becomes a robust adult with higher daily energy demands and the growth rate (hopefully) stabilizes. But even then, weight gain tends to persist, translating to a typical value for $W^{irr}/W^{rev}$ of about 0.2%. While seemingly tiny, it results in a 10 kg gain in mass, or 300 MJ of energy, over the 50 year span of a typical adult life [16].

This treatment points to the importance of considering how energy and matter are coupled in an open system such as civilization. $G$ is a total potential, so it can be decomposed into any arbitrary number of sub-components with $G = \Sigma Gi$, depending on how closely civilization

is resolved. Here we take the simplest possible approach which is to suppose, as illustrated in Fig 1, that the accessibility of energy by civilization can be defined by an interface with resources composed of $\check{n}$ material elements each with average potential $\mu$, so that

$$G(t) = \check{n}(t)\mu(t). \tag{13}$$

Thus, civilization elements are not a purely additive summation of civilization "things" $n$. Rather they represent a number of network nodes, defined in terms of people, firms, or nations that collectively enable work to dissipate energy at rate $\mathcal{E}^{sust}$.

It therefore follows from Eqs 8 and 13 that any long-term net convergence $\epsilon\mathcal{E}$ is a surplus that can be partitioned between manufacturing more civilization nodes or increasing their average potential $\mu$:

$$\epsilon\mathcal{E} = \mathcal{W}^{irr} = \frac{dG}{dt} = \mu\frac{d\check{n}}{dt} + \check{n}\frac{d\mu}{dt} \tag{14}$$

By increasing the average potential at rate $d\mu/dt$, existing civilization elements $\check{n}$ can go farther and faster. On the other hand, production of new civilization elements $d\check{n}/dt$ with average potential $\mu$ implies a phase change. Just as an excess 30 MJ of energy is required for the chemical transformation of food into a kilogram of flesh, stationary raw materials such as forests, fish stocks, and iron ore are rearranged into familiar forms such as cars, roads, and communications systems.

We assume the following thermodynamic proportionality:

$$\frac{d\mu}{d\check{n}} = (k-1)\mu\check{n} \tag{15}$$

for a constant $k$. Substituting this into Eq 14 leads to

$$\mathcal{W}^{irr} = k\mu\frac{d\check{n}}{dt} \tag{16}$$

or rearranging, net production of network nodes is related to current energy consumption through

$$\frac{d\check{n}}{dt} = \frac{\epsilon}{k\mu}\mathcal{E} \tag{17}$$

where $\epsilon/(k\mu)$ is the efficiency of converting primary energy consumption to network growth.

To summarize, we consume energy at rate $\mathcal{E}$. Most of it, namely $\mathcal{E}^{sust} = (1-\epsilon)\mathcal{E}$, is used to do reversible work at rate $\mathcal{W}^{rev}$ to sustain circulations along civilization networks. The surplus $\epsilon\mathcal{E}$ is used to do irreversible work at rate $\mathcal{W}^{irr}$ to convert raw materials into the stuff of humanity and extend existing network nodes at rate $d\check{n}/dt$. These form the fabric of society, including roads, telecommunications networks and even neural pathways encapsulating memories within our brains. We grow fastest if the efficiency $\epsilon$ is high. And through expansion of the physical interface at rate $d\check{n}/dt$ with reserves of energy and matter, civilization grows, leading recursively to further expansion and higher consumption.

## 2 Thermodynamics of global economic value

Can these strictly thermodynamic concepts be linked to the economy expressible in financial terms? Suppose for the moment that they can, and that there exists a hypothetical global quantity expressible in units of real currency (or "widgets") $W$ that expresses the size of civilization and is presumed to be proportional to energy consumption and the civilization potential

through a constant scaling factor λ:

$$\frac{G}{\tau_d} \simeq \mathcal{E} = \lambda W \qquad (18)$$

In this expression, civilization elements of whatever kind contribute to $W$ only insofar as they contribute to the overall network capacity to dissipate primary potential energy at rate $\mathcal{E}$, requiring a thermodynamic potential $G$.

From a purely dimensional perspective, the simplest possible economically quantifiable definition of $W$ is that it is an integration over time of a global quantity with units of widgets per time. The approach previously taken in [17] was to suppose that the most obvious candidate is the world economic production $Y$ (or gross domestic product GDP) calculated at market exchange rates (MER) and adjusted for inflation, in which case $W$ is the world cumulative production:

$$W = \int_0^t Y(t')dt' \qquad (19)$$

From Eqs 18 and 19, the testable hypothesis is that past economic production aggregated for all nations and integrated over all of history is tied through a constant λ to energy consumption through:

$$\mathcal{E}(t) = \lambda \int_0^t Y(t')dt' \qquad (20)$$

Note the similarity here with Eq 5. If the expression in Eq 20 can be shown to be empirically justified, it would imply that there is a constant relationship between *current* inflation-adjusted economic production $Y$ and a *rise* in global energy consumption demands $\mathcal{E}$:

$$Y = \frac{dW}{dt} = \frac{1}{\lambda}\frac{d\mathcal{E}}{dt} \qquad (21)$$

From Eqs 11 and 17, we can see that current economic production is tied to the use of primary energy to do irreversible work to convert raw materials into civilization matter. Namely,

$$Y = \frac{\mathcal{E}\eta_\mathcal{E}}{\lambda} \simeq \frac{\epsilon\mathcal{E}}{\lambda\tau_d} = \frac{k\mu}{\lambda\tau_d}\frac{d\breve{n}}{dt} = \frac{\mathcal{W}^{irr}}{\lambda\tau_d} \qquad (22)$$

In other words, with a surplus of energy $\epsilon\mathcal{E}$, the extraction of raw materials can be tied to the production of economically useful material goods (namely "widgets") as has been noted previously in [18]. These add value by speeding up human activity and increasing civilization size. Economic production provides the recipe for growth by expanding our collective interface with energy and material reserves, leading to positive increments in our capacity to consume.

Available data for testing Eq 20 include annual market exchange rate estimates of GDP, inflation-adjusted to "real" units (namely constant 2010 US dollars) from the World Bank [19] and the United Nations [20] for the years 1970 to 2017, and reconstructions from the Maddison database of the real GDP adjusted for purchasing power parity (PPP) 1990 USD for each year between 1950 and 1992, with more sparse estimates extending back to 1 CE [21] (see details in Appendix A). Annual rates of global primary energy consumption $\mathcal{E}$ are available from British Petroleum and the U.S. Department of Energy (DOE) Energy Information Administration (EIA) for the time period 1980 to 2017 [12, 22].

For initialization of the integration in Eq 19, it is estimated that world cumulative production in 1 C.E. was 250 trillion 2010 USD, a number that is obtained iteratively so that there is

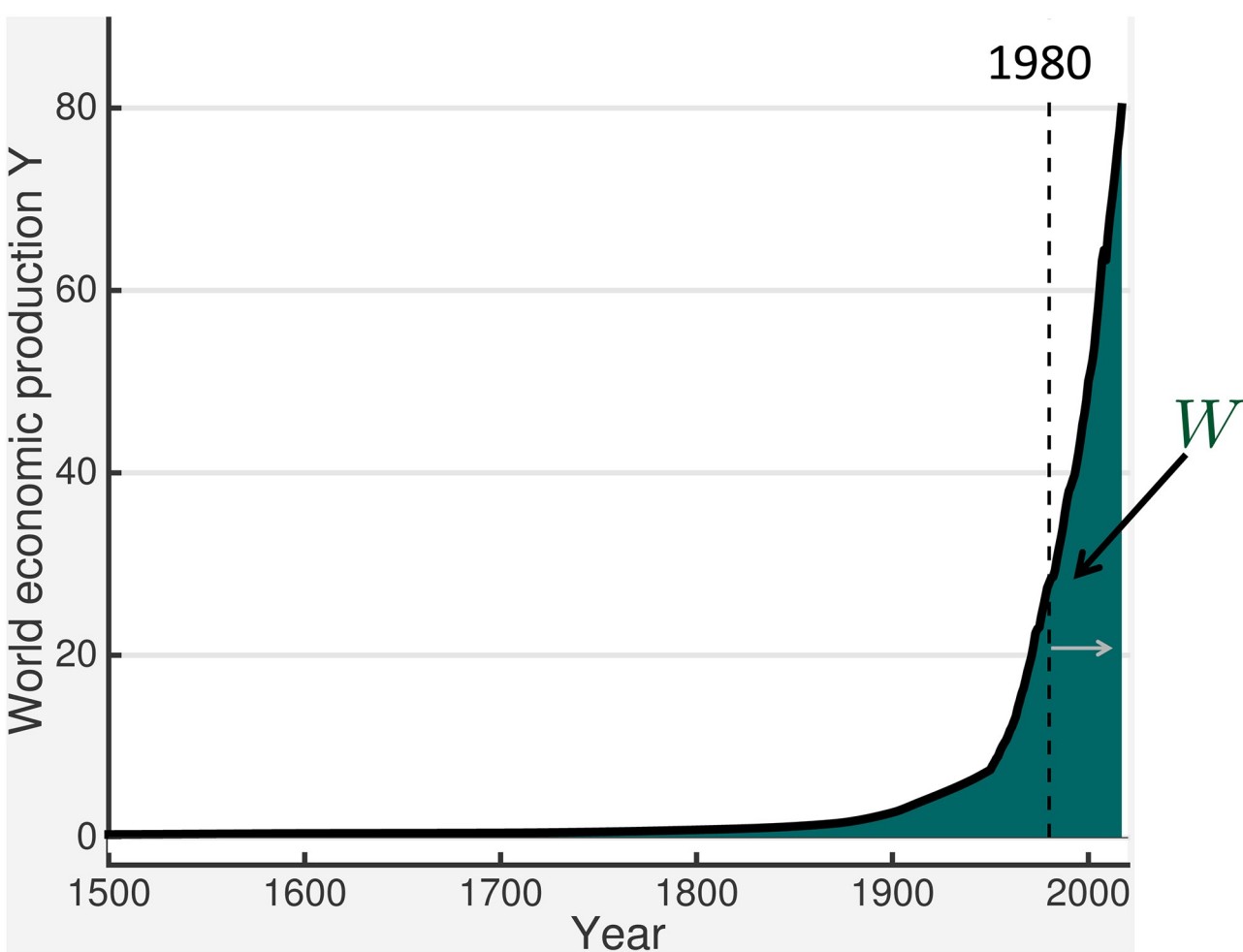

**Fig 2. Evolution of world economic production $Y$ in trillion 2010 USD per year (solid line) and the integrated contribution to the world cumulative production $W$ in trillion 2010 USD proportional to the shaded area under the curve.** The period between 1980 to 2017 that is used for comparison with world primary energy consumption $\mathcal{E}$ as described in the text is delineated by the dashed line and shown by the gray arrow.

consistency for that period between growth of $W$ and population growth rates of about 6% per century [23]. This reconstruction is about 7.3% of the value obtained for 2017, suggesting the ancient world had already evolved non-negligible wealth in its Western, Middle Eastern, and Eastern empires [24]. The Maddison database is sparse, and presumably increasingly uncertain the farther one goes back in time. However, the value of $W$ accumulated over the period 1 CE to 1000 CE is just 4.6% of the value in the year 2017. It is not until the last century that $W(t)$ grows appreciably (Fig 2). The world cumulative production between 1980 and 2017 comprises a remarkable 60% of the historically accumulated total.

The relative evolution since 1980 of $\mathcal{E}$, $Y$, $W$, $\lambda = \mathcal{E}/W$ and $\mathcal{E}/Y$ is shown in Fig 3 and summarized in Table 1. Expressing Eq 20 as a summation of yearly data, $\lambda_j = \mathcal{E}_j / \sum_0^j Y_i$, the mean value of $\lambda$ is 5.9 gigawatts per trillion 2010 USD with a standard deviation of 0.1 (2%) and an uncertainty in the mean at the 95% confidence level of 0.05 (0.8%). As an indicator of the sensitivity to uncertainty in the data, assuming for initialization of the integration a value of $W(1)$ that is double the previously derived value of 250 trillion 2010 USD, then $\lambda = 5.2 \pm 0.2$ gigawatts per trillion 2010 USD. If $W(1)$ is half as much, then $\lambda = 6.2 \pm 0.2$ gigawatts per trillion

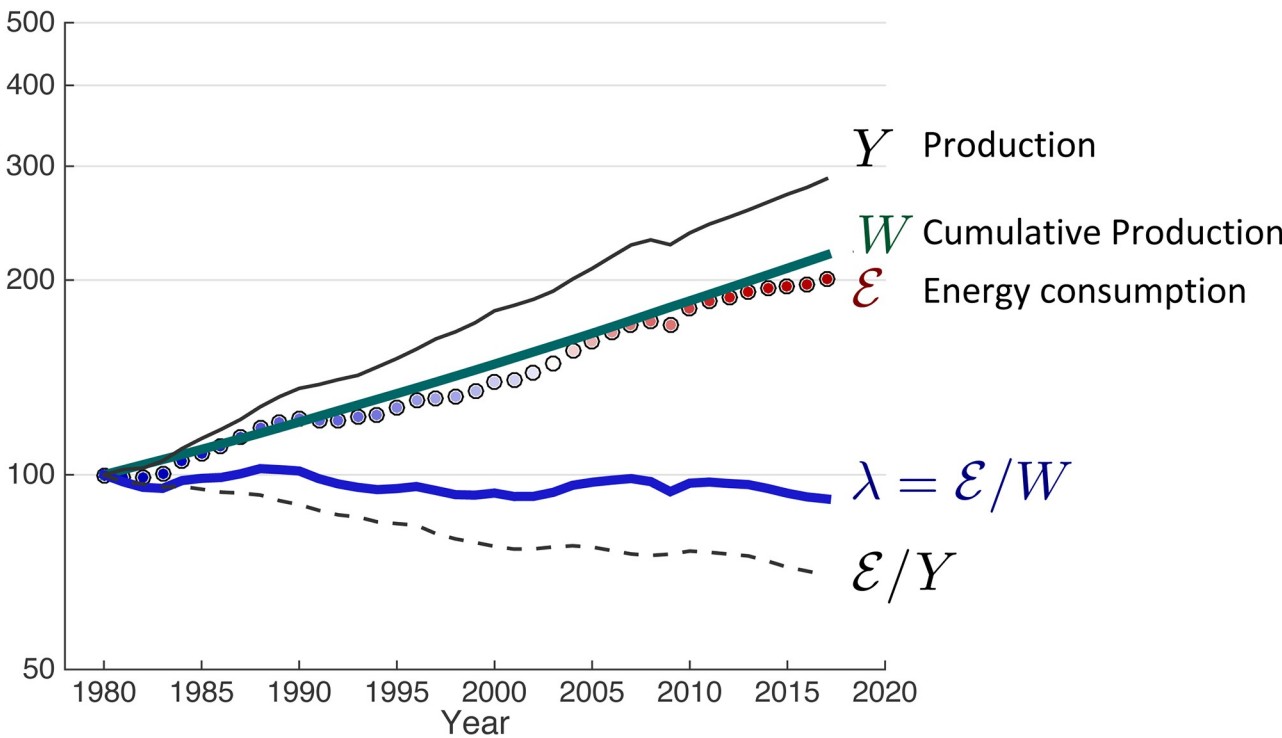

**Fig 3. Relative evolution since 1980 of the world real GDP $Y$, economic potential $W = \int_0^t Y(t')dt'$ (Eq 19), primary energy consumption $\mathcal{E}$, $\lambda = \mathcal{E}/W$ (Eq 18) and the energy intensity of production $i =: \mathcal{E}/Y = 1/\varepsilon$.**

2010 USD. While there is arguably a small secular downward trend of 0.1% yr$^{-1}$, the temporal variation in $\lambda$ is sufficiently small and insensitive to assumptions that it appears useful as a scaling factor relating an economic quantity $W$ to a physical measure $\mathcal{E}$. If $\tau_d \simeq 1$ day, then the implication is that $G/W = \lambda\tau_d \simeq 510 \pm 9$ Joules per 2010 US dollar.

The relationship in Eq 20 appears to hold empirically. But it might seem surprising given that it lacks any explicit representation of spontaneous appreciation, or of consumption, decay, and depreciation. Obviously, trade agreements and resource discovery add value without being tabulated in production. And not all of what has been produced remains as old technologies turn obsolete and networks fray due to physical destruction by storms, rust, and forgetting. A cook may produce food in a restaurant, but other than temporary body sustenance and lingering memories, the meal is gone. The total accumulated "wealth" of civilization seems difficult to relate simply to a summation of past production. Eq 20 is easier to justify, however, if there exists a steady fraction of production that contributes to civilization expansion through irreversible work, as in Eq 22.

In traditional macroeconomic growth models, there is also a quantity with units of widgets, termed real capital $K$ that relates to real economic output $Y$ according to some production function $Y = f(K, L)$ where $L$ represent labor. A common example is the Cobb-Douglas

**Table 1. The global value of $\lambda$ (gigawatts per trillion 2010 US dollar with standard deviation) defined by Eq 18 for various time periods.**

| Period | 1980-1990 | 1990-2000 | 2000-2010 | 2010-2017 | 1980-2010 | 1980-2017 |
|---|---|---|---|---|---|---|
| $\lambda = \mathcal{E}/W$ | 6.04±0.14 | 5.83±0.15 | 5.83±0.14 | 5.79±0.14 | 5.90±0.16 | 5.88±0.17 |

production function $Y = AK^\alpha L^{1-\alpha}$ where $\alpha$ is determined from a fit to data and $A$ is termed a "total factor productivity" that can itself evolve with time according to, for example, investments in research and development [25] (see also [26] for a different interpretation of $A$ in the context of a Cobb-Douglas production function taking into account energy inputs). One could argue that the appropriate theoretical relationship to draw is not between $\mathcal{E}$ and $W$ as defined in Eq 19, but between $\mathcal{E}$ and capital $K$.

To explore this possibility further, observe that capital $K$ evolves as

$$\frac{dK}{dt} = I - \delta K = Y - C - \delta K \qquad (23)$$

where $I = Y - C$ is the gross capital investment, $\delta$ is a depreciation rate, and $C$ refers to consumption of goods that are assumed to depreciate much faster than capital itself. If we define $\gamma$ as the fraction of production used for consumption plus depreciation, that is

$$\gamma Y = C + \delta K, \qquad (24)$$

then Eq 23 becomes $dK/dt = (1 - \gamma)Y$, or equivalently,

$$K(t) = \int_0^t (1 - \gamma(t'))Y(t')dt'. \qquad (25)$$

Provided $\gamma$ does not vary greatly in time, we can approximate the factor $(1 - \gamma)$ by its time average $\langle 1 - \gamma \rangle$ so that

$$K \simeq \langle 1 - \gamma \rangle \int_0^t Y(t')dt'. \qquad (26)$$

In other words, if there were a fundamental scaling between $\mathcal{E}$ and $K$, this would also imply the scaling expressed in Eq 20.

Then, if society experienced a sudden collapse in its growth toward a steady-state economy with constant $\mathcal{E}$, economic production $Y$ could remain positive, even adjusting for inflation, but the implication would be that all production would be used simply for consumption and to sustain capital $K$ at its current state, that is to say, $\gamma(t) = 1$, and both $K$ and $\mathcal{E}$ are stationary.

Unfortunately long-term global time series to test whether $K$ scales with $\mathcal{E}$ are scant, in part because there is disagreement among economists about how to appropriately aggregate the value of items as different as houses and tractors [27], and because of difficulties with estimating a starting point for the corresponding time series [28], similarly to the difficulty we mentioned above related to $W(1)$. Also, it is unclear what to include in economic capital which does not normally consider for example people or their culture, even though these are core elements of the human dissipative engine. In other words, whereas capital $K$ and the global quantity $W$ in Eq 18 may be numerically related, they are not conceptually analogous. The value expressed by $W$ lies not so much in an aggregation of inert "things", but in their summed capacity to form interconnected networks through a thermodynamic potential $G = \check{n}\mu$ that sustains the dissipation of potential energy at rate $\mathcal{E}$, whether it is from international trade, cleaning the house, or the firing of neuronal networks in human brains. A tree's past seasonal production of leaves contributes to the long term growth of its trunk, even though the leaves die each autumn. It may also be that past economic production of all kinds contributes to the cumulative growth of civilization's whole and therefore to its current energy demands.

In the meantime, even if $W$ calculated from Eq 19 yields an overestimate of the true aggregated wealth of civilization, $\lambda$ as calculated in Eq 20 has been nearly a constant for sufficiently long that it seems reasonable to assume that this regularity will carry into the future.

## 3 Implications for economic growth

Adopting a nearly fixed relationship between $W$ and $\mathcal{E}$ offers a testable basis for exploring thermodynamic constraints on economic growth. From Eqs 19 and 20, both the world cumulative production and primary energy consumption grow at the exponential rate:

$$\eta_{\mathcal{E}} = \eta_W \coloneqq \frac{1}{W}\frac{dW}{dt} = \frac{Y}{W} = \lambda\frac{Y}{\mathcal{E}} = \lambda\varepsilon \tag{27}$$

where $\varepsilon \coloneqq Y/\mathcal{E}$ is the energy efficiency of economic production or energy productivity. That is to say, higher energy productivity is related with increased energy demands.

   This result may seem counter-intuitive, but it rests only on the empirical result that $\lambda$ is a constant. It should not be confused with the concept of backfire or Jevons' Paradox [29–31], which also argues that efficiency gains lead to consumption growth, but is stated within the context of traditional economic models for specific economic sectors or nations and is often refuted. Here, Eq 27 applies strictly within a global context so that complications due to trade do not play a role. Interpreted physically, Eq 27 can be compared with Eq 22, suggesting that $\varepsilon \simeq \epsilon/\lambda\tau_d$, or that energy productivity is related to the efficiency of doing irreversible work $\mathcal{W}^{irr} = \epsilon\mathcal{E}$ that further expands the capacity to consume. The implication is that civilization is an emergent phenomenon that grew spontaneously to its currently high state of production through an energy surplus.

   The decades following 1950—known as the "great acceleration"—stand out in particular, when a relative ease of access to oil led to rapid innovations [15, 32]. To characterize such acceleration in growth, $\eta_W$ can be considered to have its own exponential growth rate

$$\eta_I \coloneqq \frac{d\ln\eta_W}{dt} \tag{28}$$

We term this an "innovation rate" because From Eq 27, if $\lambda$ is a constant, $\eta_I$ can also be expressed as

$$\eta_I = \eta_{\varepsilon} \coloneqq \frac{d\ln\varepsilon}{dt} \tag{29}$$

Whenever conditions support an increase in energy productivity $\varepsilon$ (or thermodynamic efficiency $\epsilon$), for example due to energy reserve discoveries [15], then $\eta_I > 0$ and civilization growth is superexponential.

   The world GDP growth rate $\eta_Y \coloneqq d\ln Y/dt$ follows a slightly different pathway. From $Y = \varepsilon\mathcal{E}$ and Eq 27, it follows that:

$$\eta_Y = \eta_{\mathcal{E}} + \eta_{\varepsilon} = \lambda\varepsilon + \eta_{\varepsilon} \tag{30}$$

At global levels, by reducing the energy required for manufacture of valuable goods, civilization bootstraps through innovation to higher GDP growth and accelerating rates of energy consumption.

   Table 2 shows a check of the applicability of the derived growth Eqs 27 to 30 for the period between 1980 and 2010 and for a shorter more recent period from 2010 to 2017. Overall, there is close agreement between observations and calculated rates of change based on the constancy of $\lambda$, which suggests the relationship offers a useful tool for making simplified predictions of economic growth. However, there are also some discrepancies after 2010. For example, as stated in Eq 27, a consequence of a constant value for $\lambda$ is that the primary energy consumption growth rate $\eta_{\mathcal{E}}$ should be equivalent to both the growth rate of the world cumulative

**Table 2. Measured average growth rates (%/yr) compared with rates derived assuming λ is a constant in bold.** Pertinent equations are in parentheses.

| | Wealth | Energy | | Innovation | | GDP | |
|---|---|---|---|---|---|---|---|
| | $\eta_W$ | $\eta_{\mathcal{E}}$ | $\lambda\varepsilon$ (27) | $\eta_I$ (28) | $\eta_{\varepsilon}$(29) | $\eta_Y$ | $\lambda\varepsilon + \eta_{\varepsilon}$ (30) |
| 1980-2010 | 2.06 | 1.98 | **2.09** | 0.82 | **0.91** | 2.88 | **3.0** |
| 2010-2017 | 2.33 | 1.60 | **2.40** | 0.45 | **1.18** | 2.78 | **3.58** |
| 1980-2017 | 2.14 | 1.84 | **2.15** | 0.73 | **1.0** | 2.84 | **3.15** |

production $\eta_W$ and the product $\lambda\varepsilon$. For the period 1980 to 2010 over which energy consumption increased by 80%, the three calculations differ by at most 6%.

Similarly the GDP growth rate calculated from the expression $\lambda\varepsilon + \eta_{\varepsilon}$ (rightmost in Eq 30) agrees to within 4% with the directly calculated value $\eta_Y$. However, for the shorter period between 2010 and 2017, the agreement is less precise, and also for the time period 1980 to 2017. The reason is unknown, although note from Table 2 and the mathematical equivalency $\eta_Y = \eta_{\mathcal{E}} + \eta_{\varepsilon}$ that any discrepancy between $\eta_Y$ and $\lambda\varepsilon + \eta_{\varepsilon}$ is due entirely to discrepancies between $\eta_{\mathcal{E}}$ and $\lambda\varepsilon$. Energy consumption increased by just 11% between 2010 and 2017 so there is greater susceptibility to quantification errors. Also, nominal GDP is what is measured, and any calculation of real $Y$ requires an accurate assessment of the GDP deflator that attempts to account for inflation, and its true magnitude may have been underestimated. Indeed, $\lambda = \mathcal{E}/W$ was 2% lower for the period between 2010 and 2017 (Table 1), and any departure from constancy affects calculation of higher order derivatives such as the growth rates $\eta_{\mathcal{E}}$ and $\eta_Y$.

## 4 Implications for carbon dioxide emissions and concentrations

### 4.1 Simplifications to socio-economic drivers

The anthropogenic contribution to atmospheric carbon dioxide emissions can be conveniently decomposed into the product of population $P$, affluence expressible as the gross domestic product (GDP) per person $g := Y/P$, the energy intensity of economic production $i =: \mathcal{E}/Y = 1\varepsilon$, and the amount of carbon dioxide emitted by the choice of energy source $c =: C/\mathcal{E}$, leading to the Kaya identity $C = P \times g \times i \times c$ [33]. In terms of growth rates, we have

$$\eta_C = \eta_P + \eta_g - \eta_\varepsilon + \eta_c \tag{31}$$

The most recent IPPC report [34] lists rapid increases in population and standard of living as primary drivers of past emissions growth but focuses on innovations that improve production efficiency and reductions in the carbon intensity of fuels as targets for future reductions.

Eq 31 helps frame issues surrounding climate change mitigation. But it is only a mathematical identity and, as such, it does not directly allow for dynamic interactions between terms. What the link expressed in Eq 20 between current consumption and the economy provides is an added strong constraint on interrelationships between population, standard of living, and production efficiency. For example, inserting Eq 20 in $C = c\mathcal{E}$ yields the result that current carbon dioxide emissions can be related to <u>past</u> accumulated economic production through

$$C(t) = \lambda c \int_0^t Y(t')dt' \tag{32}$$

A revised expression then follows for Eq 31, namely

$$\eta_C = \eta_c + \eta_\mathcal{E} = \eta_c + \lambda\varepsilon \tag{33}$$

where we used Eq 27.

**Table 3. Average growth rates in carbonization and CO$_2$ emissions (%/yr).** Rates derived assuming λ is a constant are shown in bold, and pertinent equations in parentheses. The units of $C/W$ are Gt C yr$^{-1}$ per quadrillion 2010 USD.

| | Scaling (Eq 32) | Carbonization $c = C/\mathcal{E}$ | CO$_2$ emissions $C$ | |
| --- | --- | --- | --- | --- |
| | $C/W = \lambda c$ (±std. dev.) | $\eta_c$ | $\eta_C$ | $\eta_c + \lambda\varepsilon$ (33) |
| 1980-2010 | 1.50±0.06 | -0.21 | 1.77 | **1.88** |
| 2010-2017 | 1.45±0.05 | -0.36 | 1.25 | **2.04** |
| 1980-2017 | 1.49±0.06 | -0.25 | 1.59 | **1.90** |

A test of Eq 33 is shown in Table 3, based on the values for $\lambda\varepsilon$ shown in Table 2 and data from the Global Carbon Atlas [35]. Between 1980 and 2010, the observed average annual rate of emissions growth $\eta_C$ was 1.77% yr$^{-1}$, within 6% of the calculate value of 1.88% yr$^{-1}$ for $\eta_c + \lambda\varepsilon$. For more recent years, the discrepancy is 20%.

Similarly, the revised identity for CO$_2$ emissions given by Eq 33 implies that

$$\eta_P + \eta_g = \lambda\varepsilon + \eta_\varepsilon \tag{34}$$

A test of this relation is shown in Table 4. For the 1980 to 2017 time period, the difference between both sides of the equality in Eq 34 is about 10%. Also, note that over the longer term $\eta_p$ and $\eta_g$ are remarkably similar. If $\breve{n}$ can be related to population and $\mu$ to standard of living, this empirical result is consistent with the thermodynamic relationship given by Eq 15 with $k = 2$.

As discussed, the production efficiency can be related to the energy efficiency through $\varepsilon \simeq \epsilon/(\lambda\tau_d)$. This suggests that, whether it is CO$_2$ emissions, population, or standard of living, it is the fractional imbalance between energy consumption and dissipation that drives growth. A surplus enables irreversible work to be done to make more of everything, including people, and speeding it all up. Moreover, current efficiency levels arose from an accumulation of prior innovations:

$$\varepsilon = \int_0^t \frac{d\varepsilon(t')}{dt'} dt' \tag{35}$$

so a conclusion might be reached that it is current and past improvements in production efficiency that have driven current growth in emissions, population, and standard of living. Population and emissions growth rates have inertia because the world has memory of its past innovations.

## 4.2 Emissions stabilization and climate change mitigation

So what can be done to reverse the course of growing CO$_2$ emissions? Eq 33 suggests that stabilizing carbon dioxide emissions will require the economy to decarbonize at a rate $\eta_c = -\lambda\varepsilon$ that

**Table 4. Average growth rates of population and standard of living (%/yr).** Summed rate derived assuming λ is a constant from efficiency estimates are shown in bold.

| | Population | Standard of living | Summation | |
| --- | --- | --- | --- | --- |
| | $\eta_P$ | $\eta_g$ | $\eta_p + \eta_g$ | $\lambda\varepsilon + \eta_\varepsilon$ (34) |
| 1980-2010 | 1.45 | 1.43 | 2.88 | **3.0** |
| 2010-2017 | 1.10 | 1.68 | 2.78 | **3.58** |
| 1980-2017 | 1.38 | 1.46 | 2.84 | **3.15** |

is as fast as the rate of energy consumption growth $\eta_\varepsilon = \lambda \varepsilon$. In the period 2010-2017, we observed $\lambda \varepsilon$ to be about 2.4% per year (Table 2). For a sense of what this implies, consider that 2.4% of the current global rate of energy consumption $\mathcal{E} \approx 20$ TW corresponds to 480 GW. That is to say, for energy consumption to continue to grow at this rate without increasing carbon dioxide emissions would require over 1 GW of new power capacity in nuclear or renewable energy to be added online each day, the approximate size of a large central power plant. The corresponding figure using $\eta_\varepsilon = 1.6\%$ directly (Table 2), instead of the value $\lambda \varepsilon = 2.4\%$, is 320 GW, or just under 1 GW of new power capacity in renewable energy per day.

Furthermore, stabilized emissions do not lead to stabilized atmospheric $CO_2$ concentrations, not until there is a balance with natural uptake by the land and oceans. As a crude approximation, land and ocean sinks are linearly proportional to the perturbation $\Delta[CO_2]$ from pre-industrial concentrations of approximately 275 ppmv [36]. More sophisticated approaches are possible that permit accurate projections over multi-century timescales [37].

Nonetheless, a simple linear model readily lends insights into forces driving near term evolution while remaining consistent with multiple decades of observations. Data show that the linear sink rate $\sigma$ to the land and oceans with respect to the decadally averaged perturbation $\Delta[CO_2]$ was 2.3 ± 0.5% $yr^{-1}$ in the 1980s, 2.4 ± 0.4% $yr^{-1}$ in the 1990s and 2.2 ± 0.4% $yr^{-1}$ in the 2000s [38]. There is no inter-decadal trend and the uncertainty is greater than the variability. Thus, an average value of 2.3 ± 0.4% $yr^{-1}$ is assumed here.

The effect of carbon emissions on atmospheric $CO_2$ concentrations can be obtained by normalizing by the atmospheric mass. Every gigaton of emitted carbon corresponds to $\kappa = 0.47$ parts per million by volume (ppmv) of increased $CO_2$ concentration [39]. The approximate balance equation is then

$$\frac{d\Delta[CO_2]}{dt} = \kappa C - \sigma \Delta[CO_2]. \tag{36}$$

Substituting Eq 32, carbon dioxide emissions can then be related to past economic production and the carbonization of the energy source $c$ through the integro-differential equation

$$\frac{d\Delta[CO_2]}{dt} = \kappa \lambda c \int_0^t Y(t')dt' - \sigma \Delta[CO_2] \tag{37}$$

Taking $\kappa$ and $\lambda$ as constants, then stabilization or reduction of concentrations at any given perturbation value $\Delta[CO_2]$ requires the following limits on the carbonization of emissions

$$c \leq \frac{\sigma}{\kappa \lambda} \frac{\Delta[CO_2]}{\int_0^t Y(t')dt'} \simeq 0.26 \frac{\Delta[CO_2]}{W} \tag{38}$$

where $c$ has units of Gt C $EJ^{-1}$, $W$ has units of trillion 2010 USD and the numerical coefficient $\sigma/(\kappa \lambda) \simeq 0.26$ has units of (Gt C) × (trillion 2010 USD) × (ppmv$^{-1}$) × ($EJ^{-1}$).

Recent values for $c$ are close to 0.017 Gt C $EJ^{-1}$, so despite a recent surge in renewables, $c$ is not rapidly declining [40]. As shown in Table 3, the annual decarbonization rate $\eta_c$ in recent years is just 0.36% $yr^{-1}$ and the correspondence between $CO_2$ emissions $C$ and cumulative global production $W$ expressed as $C/W = \lambda c$ has effectively been unchanged over the past four decades. Based on these observations, and to develop an intuition for constraints on the future, we take the baseline assumption that $\lambda c$ will remain fixed. In this case, Eq 37 implies a straightforward proportionality relating the world cumulative production $W$ and the equilibrium value for the concentration perturbation $\Delta[CO_2]_{eq}$. Setting $d\Delta[CO_2]/dt = 0$ it follows that:

$$W = \frac{\sigma}{\kappa \lambda c} \Delta[CO_2]_{eq} \simeq 15.4 \Delta[CO_2]_{eq} \tag{39}$$

**Table 5. Average values of the scaling $W/\Delta\,[CO_2]_{eq} = \sigma/(\kappa c \lambda)$ defined by Eq 39 for various time periods, in trillion 2010 USD ppmv$^{-1}$.**

| Period | 1980-1990 | 1990-2000 | 2000-2010 | 1980-2010 |
|---|---|---|---|---|
| $\sigma/(\kappa c \lambda)$ | 15.0 | 16.3 | 14.9 | 15.4 |

Here, the numerical coefficient $\sigma/(\kappa c \lambda) \simeq 15.4$ is obtained from the years 1980 to 2010 and has units of trillion 2010 USD ppmv$^{-1}$. Values of this coefficient for different periods are shown in Table 5.

Comparing Eq 39 with Eq 18 we obtain

$$\Delta[CO_2]_{eq} \simeq \frac{\kappa c}{\sigma \tau_d} G$$

Effectively, at equilibrium with land and ocean sinks, civilization's combustion garbage heap in the form of an atmospheric $CO_2$ perturbation is linearly proportional to how far thermodynamically it has stretched itself away from the environmental base state of $G = 0$. Both rise as our collective historical achievement.

Fig 4 shows the relationship between the world cumulative production $W$ and $CO_2$ concentrations, both for the past 2000 years and for the future, assuming no future decarbonization and that energy consumption persists at its current growth trend of 2.4% yr$^{-1}$, a conservative estimate perhaps given that it implies an equivalent rate of world GDP growth (Eq 30). That is to say, the open circles for each future date correspond to the non-equilibrium values for $CO_2$ concentrations derived from Eq 37 assuming that $W$ grows at the constant rate $\eta_W = \lambda \varepsilon$. In addition, for each value of $W$ the set of solutions is provided for the $CO_2$ concentration at which emissions stabilize with land and ocean sinks by applying Eq 39, namely the equilibrium concentration to which civilization is committed even for the mathematically extreme case that $W$ were to remain constant (in other words, not only zero GDP growth, but effectively zero inflation-adjusted production).

The equilibrium $CO_2$ concentration is approached asymptotically with timescale $\tau_{CO_2} = 1/\sigma$, so that the difference is halved in about 30 years. For example, Fig 4 shows that stabilizing concentrations at a nominal value of 350 ppm would require that the current world cumulative production shrink by two thirds to a value not seen since 1960.

It is probably safe to assume that civilization will not willingly engage in such drastic pruning. Looking to the future, Fig 4 shows that without rapid decarbonization, we have already committed ourselves to $CO_2$ concentrations above 500 ppmv, well in excess of the 450 ppmv threshold that has been deemed "dangerous" [44]. At current growth rates, the commitment is to a doubling of pre-industrial levels by 2030, and to eventual levels close to 650 ppmv by 2040.

It should also be noted that $CO_2$ uptake is not in fact linear over timescales much longer than decades. The values for $[CO_2]_{eq}$ presented here do not reflect important non-linearities that might arise from e.g. increasing ocean acidification, and that would allow for concentrations to continue to slowly rise even as energy consumption rates stall [37]. With respect to globally-averaged surface temperature anomalies, it has been argued that they have a linear relationship with cumulative emissions of carbon. This sensitivity depends only weakly upon whether emissions are rising and falling, and the maximum $CO_2$ concentration that is reached. A value of 1.6°C per Tt C [45] can be used as a rough guide, in which case persistence of current 10 Gt C yr$^{-1}$ emissions rates would imply a further temperature rise of about 0.5°C by 2050. Assuming persistence in current 2.4% yr$^{-1}$ energy consumption growth rates, and no further decarbonization, the increase is 0.7°C.

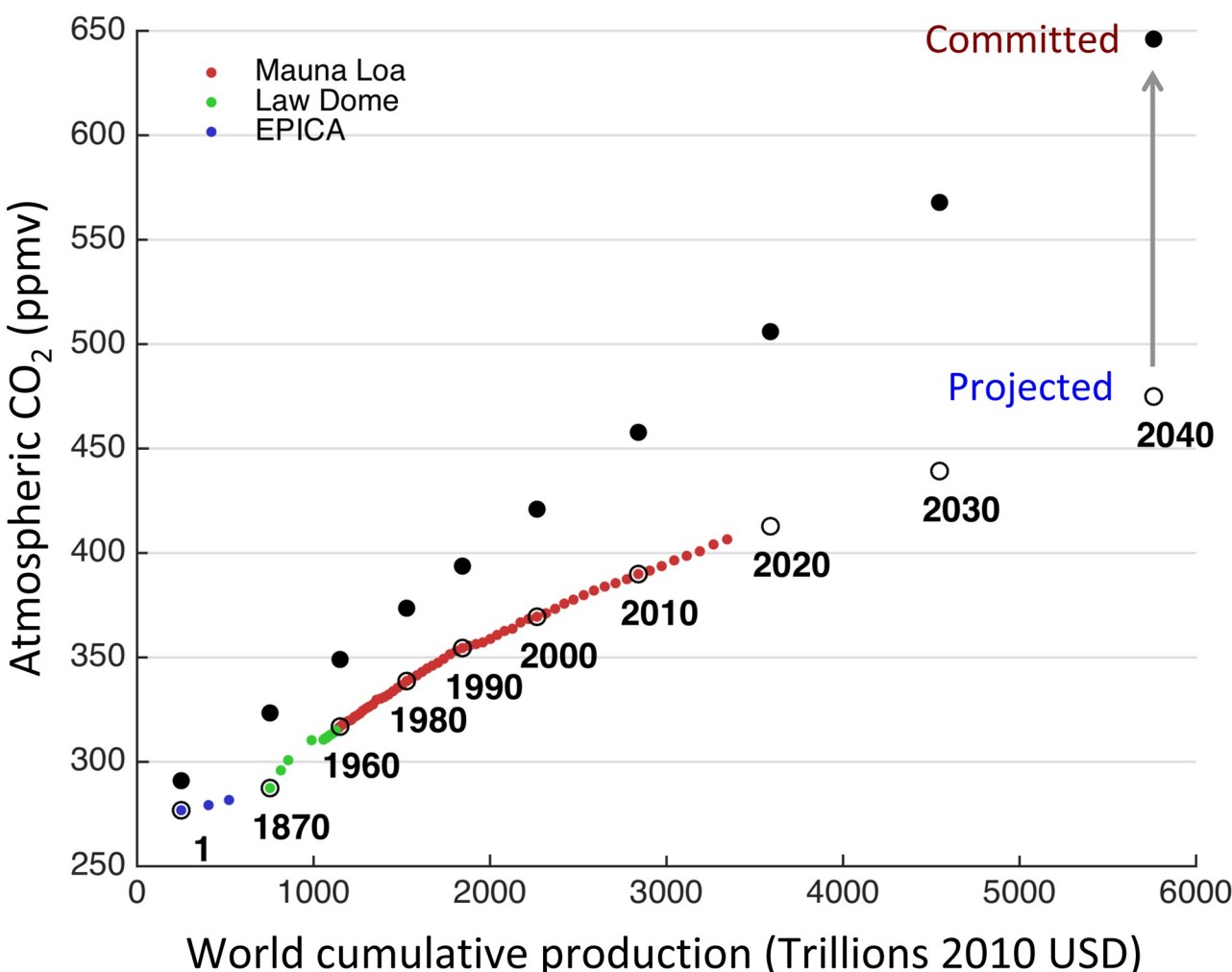

**Fig 4. Historical reconstructions of world cumulative production** $W$ **and atmospheric $CO_2$ concentrations with projections assuming** $\eta_c = 0$ **and** $\eta_\varepsilon = \eta_\varepsilon$ **(2017) = 2.4%yr$^{-1}$ (circles) and corresponding stabilization concentrations from Eq 39.** The halving time between predicted and committed is about 30 years. Concentration data includes flask samples from Mauna Loa [41] and Antarctic ice core data [42, 43].

## 5 Conclusions

This article identifies a persistent relationship between global energy consumption and cumulative economic production. It implies that a surprisingly simple description of the human system is sufficient to explain past global trends and make robust projections of the aggregated world economy and its waste products. Humanity grows when more energy is available than it requires for its daily needs. Then work can be done not just for sustenance but for expansion. Because current sustenance demands emerge from past growth, inertia plays a much more important role in determining future societal and climate trajectories than has been generally acknowledged, particularly in the physically unconstrained models that are widely used to link the economy to climate [46, 47]. We have accumulated over history a long series of innovations in efficiency that continue to propel us forward. Without forgetting these advances, we will maintain a continued ability to expand our interface with the primary resources we consume.

Eventually, of course, the interwoven networks of civilization will unravel and emissions will decline, whether it is through depletion of resources, environmentally forced decay or—as demonstrated recently—pandemics [48]. But the cuts will have to be deep, continuous, and cumulative to overcome the tremendous accumulated growth we have sustained up to this point.

The formulations presented here are intended to help constrain the problem by reducing the number of available targets that can reasonably be expected to lead to avoidance of extreme climate change. Notably, gains in energy efficiency play a critical role in enabling increases in population and prosperity, and in turn growth of energy demands and carbon dioxide emissions, contrary to what would reasonably be assumed if civilization did not grow [33, 49, 50]. What seems to be required is a peculiar dance between reducing the production efficiency of civilization while simultaneously innovating new technologies that move us away from combustion.

The relationships identified all stem mathematically from the falsifiable identity $\mathcal{E}(t) = \lambda W(t)$, where $W(t) = \int_0^t Y(t')dt'$. While the specific value of $\lambda$ that was identified is 5.9±0.2 gigawatts per trillion 2010 US dollars, what matters from the standpoint of calculating trends is that the ratio $\mathcal{E}(t)/\int_0^t Y(t')dt'$ is a constant to within observational uncertainty. Further theoretical work is required to link the relationship to more traditional macroeconomic modeling frameworks. Continued observations will provide a useful check on its validity. Any evidence of a sustained downward trend in $\lambda$ may help pinpoint decoupling of economic production from civilization's metabolic needs.

## A Appendix: Calculation of cumulative production

Market exchange rate estimates of $Y_i$, inflation-adjusted to "real" constant year 2010 dollars, are available from the World Bank and the United Nations for the years between 1970 and 2017 [19, 20]. Estimates of real GDP adjusted for purchasing power parity (PPP) 1990 USD are available for each year between 1950 and 1992, and in larger intervals extending back to 1 CE [21]. To calculate $W$ these estimates are converted to market exchange rate MER inflation-adjusted 2010 values. For the time period between 1970 and 1992 for which concurrent MER and PPP statistics are available, the mean inflation-adjusted ratio PPP/MER is $\kappa = 1.205$ with no clear trend.

A historical reconstruction of the annual global GDP is obtained by dividing the Maddison PPP values by $\kappa$ between 1 C.E. and 1970 C.E, applying a cubic spline between sparse data points to obtain annual values, and using World Bank statistics for more recent years [19]. The value of world cumulative production $W$ is then

$$W(t) = W(1) + \sum_1^t Y(t) \tag{40}$$

where $W(1)$ refers to total accumulated world cumulative production to date in 1 C.E. To obtain a value for $W(1)$, it is assumed that $W$ and world population grew equally fast at that time. Available statistics suggest a population in ca. 1 C.E. [23] that was 170 million and growing by 10 million every hundred years, at a rate of $\eta_{pop} = 0.059\%$ per year. The estimated value for the real MER GDP in 1 C.E. is 0.147 trillion 2010 USD. Assuming that civilization population and wealth grew at the same rate, i.e., $\eta_{pop} = \eta_W$, then from Eq 19 it follows that $W(1) = 250$ trillion 2010 USD.

One criticism might be that MER dollars should be adjusted to PPP dollars [51] since market exchange rates fail to account for differences in how people in different countries value

equivalent baskets of goods. One rebuttal has been that such equivalents do not exist because different cultures value goods differently and that any discrepancies tend to diminish over time with a half life of three to five years due to the pressures of international and domestic trade [52]. In the case of the work here, there is another counter-argument which is that there is no intent to address short-term inequalities between nations, only the global sum of all of civilization and its evolution over they long-run. Effectively, there is only one "basket of goods", and that is humanity taken as a whole, including all its social and physical networks.

Rates of global primary energy consumption from all sources $\mathcal{E}$ are available from the U.S. Department of Energy (DOE) Energy Information Administration (EIA) for the time period 1980 to 2016 and from British Petroleum between 1965 and 2017 [12, 22]. Rates of global primary energy consumption and production provided by the EIA have a mean ratio of 99.83% so here it is assumed that the two are equivalent.

Data is available in Supplementary Materials.

## Supporting information

**S1 Table. Energy, production, and emissions statistics since 1980.**
(XLSX)

**S2 Table. Reconstructions of energy, production, cumulative production, and population since 1 CE.**
(XLSX)

## Author Contributions

**Conceptualization:** Timothy J. Garrett, Matheus Grasselli, Stephen Keen.

**Data curation:** Timothy J. Garrett.

**Formal analysis:** Timothy J. Garrett.

**Funding acquisition:** Stephen Keen.

**Investigation:** Timothy J. Garrett, Matheus Grasselli.

**Methodology:** Timothy J. Garrett, Matheus Grasselli.

**Validation:** Timothy J. Garrett.

**Visualization:** Timothy J. Garrett.

**Writing – original draft:** Timothy J. Garrett, Matheus Grasselli.

**Writing – review & editing:** Timothy J. Garrett, Matheus Grasselli, Stephen Keen.

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
