## [Decision Letter · Decision Letter 0]

13 Jul 2020

PONE-D-20-17177

Past production constrains current energy demands: persistent scaling in global energy consumption and implications for climate change mitigation

PLOS ONE

Dear Dr. Garrett,

Thank you for submitting your manuscript to PLOS ONE. After careful consideration, we feel that it has merit but does not fully meet PLOS ONE’s publication criteria as it currently stands. Therefore, we invite you to submit a revised version of the manuscript that addresses the points raised during the review process.

Please see comments below.

We look forward to receiving your revised manuscript.

Kind regards,

Omeid Rahmani, Ph.D

Academic Editor

PLOS ONE

Additional Editor Comments:

The authors should follow the order of Eq. initially. From Fig. 1 caption, there is a disorder in the numbering of equations.

Reviewers' comments:

Reviewer's Responses to Questions

**Comments to the Author**

1. Is the manuscript technically sound, and do the data support the conclusions?

Reviewer #1: Yes

Reviewer #2: Yes

2. Has the statistical analysis been performed appropriately and rigorously? 

Reviewer #1: Yes

Reviewer #2: Yes

3. Have the authors made all data underlying the findings in their manuscript fully available?

Reviewer #1: Yes

Reviewer #2: Yes

4. Is the manuscript presented in an intelligible fashion and written in standard English?

Reviewer #1: Yes

Reviewer #2: Yes

5. Review Comments to the Author

Reviewer #1: This research is a professionally-written interesting piece to read with credible data sources presenting deep knowledge of the authors.

First comment: According to the authors, past production determines the level of energy demand. However, inside the text I have not seen any proof/justification to show the direction of causation. To elaborate more, how one can say that the causation does not run in the opposite direction, i.e., from energy demand to production.

Second comment: Another clarification that I would like to see is that according to the authors "there is persistent time-independent scaling between the historical time integral of world inflation-adjusted economic production and current rates of world primary energy consumption". To my understanding, historical time-integral is a time-dependent concept which contradicts the "time-independent" claim by the authors.

Last comment: for the model developed by the authors, is it possible to account for other factors such as governmental and/or international intervention policies/strategies towards decarbonization?

Reviewer #2: Authors of the article “Past production constrains current energy demands: persistent scaling in global energy consumption and implications for climate change mitigation” have selected an interesting topic and developed it well. In my opinion, the paper is qualified to publish in the current form.

6. PLOS authors have the option to publish the peer review history of their article (what does this mean?). If published, this will include your full peer review and any attached files.

Reviewer #1: **Yes: **Reza Ghazal

Reviewer #2: No

---

## [Author Response · Author response to Decision Letter 0]

28 Jul 2020

See attached file for response to reviewers

---

## [Editor Report · Decision Letter 1]

31 Jul 2020

Past world economic production constrains current energy demands: persistent scaling with implications for economic growth and climate change mitigation

PONE-D-20-17177R1

Dear Dr. Garrett, 

We’re pleased to inform you that your manuscript has been judged scientifically suitable for publication and will be formally accepted for publication once it meets all outstanding technical requirements.

Kind regards,

Omeid Rahmani

Academic Editor

PLOS ONE

Additional Editor Comments (optional):

Please check the order of equations especially from the caption of Figure 1.
---

## [Editor Report · Acceptance letter]

18 Aug 2020

PONE-D-20-17177R1 

Past world economic production constrains current energy demands: persistent scaling with implications for economic growth and climate change mitigation 

Dear Dr. Garrett:

I'm pleased to inform you that your manuscript has been deemed suitable for publication in PLOS ONE. Congratulations! Your manuscript is now with our production department. 

Kind regards, 

on behalf of

Dr. Omeid Rahmani 

Academic Editor

PLOS ONE